# Clinical Management in Traumatic Brain Injury

**DOI:** 10.3390/biomedicines12040781

**Published:** 2024-04-02

**Authors:** Amy Yan, Andrew Torpey, Erin Morrisroe, Wesam Andraous, Ana Costa, Sergio Bergese

**Affiliations:** 1Department of Anesthesiology, Renaissance School of Medicine, Stony Brook University, Stony Brook, NY 11794, USA; amy.yan@stonybrookmedicine.edu (A.Y.); andrew.torpey@stonybrookmedicine.edu (A.T.); wesam.andraous@stonybrookmedicine.edu (W.A.); ana.costa@stonybrookmedicine.edu (A.C.); 2Renaissance School of Medicine, Stony Brook University, Stony Brook, NY 11794, USA; erin.morrisroe@stonybrookmedicine.edu; 3Department of Anesthesiology and Neurological Surgery, Renaissance School of Medicine, Stony Brook University, Stony Brook, NY 11794, USA

**Keywords:** traumatic brain injury, intracranial pressure, intracranial hypertension, cerebral blood flow, brain, head, trauma

## Abstract

Traumatic brain injury is one of the leading causes of morbidity and mortality worldwide and is one of the major public healthcare burdens in the US, with millions of patients suffering from the traumatic brain injury itself (approximately 1.6 million/year) or its repercussions (2–6 million patients with disabilities). The severity of traumatic brain injury can range from mild transient neurological dysfunction or impairment to severe profound disability that leaves patients completely non-functional. Indications for treatment differ based on the injury’s severity, but one of the goals of early treatment is to prevent secondary brain injury. Hemodynamic stability, monitoring and treatment of intracranial pressure, maintenance of cerebral perfusion pressure, support of adequate oxygenation and ventilation, administration of hyperosmolar agents and/or sedatives, nutritional support, and seizure prophylaxis are the mainstays of medical treatment for severe traumatic brain injury. Surgical management options include decompressive craniectomy or cerebrospinal fluid drainage via the insertion of an external ventricular drain. Several emerging treatment modalities are being investigated, such as anti-excitotoxic agents, anti-ischemic and cerebral dysregulation agents, S100B protein, erythropoietin, endogenous neuroprotectors, anti-inflammatory agents, and stem cell and neuronal restoration agents, among others.

## 1. Introduction

According to the Global Burden of Disease Study, there were 27.08 million new cases of traumatic brain injury (TBI) worldwide in 2016 [1]. In the United States, more than one million patients seek medical attention for TBIs, with approximately 50,000 deaths yearly and 5.3 million people living with long-term TBI-related disabilities [2]. In the European Union, 57,000 deaths and 1.5 million hospital admissions are attributed to TBI [3]. Children and adults over 65 years of age are most likely to sustain a TBI. Falls are the leading causes of TBIs, while motor vehicle accidents are the leading cause of death from TBIs. The estimated annual cost of TBIs (including direct and indirect costs) in 2010 is USD 76.56 billion [4]. Consequences stemming from TBI include memory impairment, mood disorders, and physical disability [5]. 

## 2. Mechanisms of Brain Injury during Trauma

Mechanisms of TBI are classified as: (1) focal brain damage due to a contact injury causing laceration, contusion, and/or intracranial hemorrhage; (2) diffuse brain damage from acceleration/deceleration injury leading to brain swelling and/or diffuse axonal injury [6,7]. The mechanisms of injury can be divided into primary injury (primary/mechanical damage attributed to injury at the time of trauma) and secondary injury (secondary damage, delayed non-mechanical damage that occurs later as a result of a complication from the initial trauma). Primary injuries include skull fractures, intracranial hemorrhage, and diffuse axonal injury. Secondary injury includes cerebral edema, increase in intracranial pressure, neurotransmitter changes, inflammation, hypoperfusion/hyperperfusion, and cerebral ischemia [4]. Primary injuries are typically sensitive to preventive but not therapeutic treatments, whereas secondary injuries such as intracranial hypertension and cerebral ischemia are responsive to therapeutic measures [6]. 

### 2.1. Primary Injury

Skull fractures can be differentiated based on location (flat bones versus skull base), appearance (linear versus comminuted), degree of depression, and degree of communication. Fractures have a greater chance of posttraumatic seizures and infection, as fractured depressed bone can penetrate dura or tissue. Open skull fractures result in direct communication with the scalp or respiratory mucosa, leading to an increased risk of central nervous system infection [8]. If an open skull fracture results in extracranial cerebral spinal fluid (CSF) leakage from the nose or ears, then bacteria and air can enter the skull, producing pneumocephalus or infectious processes like brain abscess or meningitis. The risk of posttraumatic meningitis may increase if patients undergo burr holes in the emergency department, decompressive craniectomy, craniotomy, and/or repeat surgery within 48 h [9]. 

Damage to blood vessels within the brain causes intracranial hemorrhage, and the location of bleeding results in different types of intracranial hematomas. An epidural hematoma results from the laceration of the dural veins or arteries between the dura and skull area. A common source of bleeding in epidural hematomas is the middle meningeal artery after the fracture of a temporal bone. Epidural hematomas originating from an arterial source often result in instant neurological deterioration as there is a rapid accumulation of blood that exerts greater pressure [10]. A subdural hematoma often occurs in the subdural place in severe TBI. The accumulation of blood is limited within the arachnoid membrane and dura, so the hematoma does not develop as rapidly, but this leads to mass lesions, leading to an estimated rate of mortality of 60–80% [11]. Subarachnoid hemorrhages result from accumulation of blood between the arachnoid and pia mater [12]. 

In many TBIs, the brain can be injured in a coup–contrecoup mechanism, where there is a brain contusion at both the initial site and the opposite side of the insult, due to movement of the brain within the skull. Cerebral contusions often originate from the frontal or temporal poles. The energy of the insult leads to the rupture of micro vessels, which cause extravasation of blood and the inability of these vessels to perfuse tissues. The breakdown of the blood and resultant ischemia from this damage contribute to secondary injury [13]. 

At the cellular level, the primary injury results in the rupture and necrotic death of neurons, astrocytes, and oligodendrocytes, as well as the disruption of the neuronal interconnection. This disruption, called a diffuse axonal injury, is classified by the Adams classification. Grade 1 refers to mild diffuse external injury with microscopic white matter changes of the cerebral cortex, corpus callosum, and brain stem. Grade 2 describes moderate diffuse axonal injury with focal lesions in the corpus callosum. Grade 3 includes the lesions in Grade 2 as well as additional focal lesions in the brain stem. When the cytoskeletons of axons are disrupted, it results in proteolysis, swelling, and molecular changes in neuronal structure and function [14]. Coup–contrecoup injuries lead to a significantly higher mortality in patients less than 60 years of age and in patients with a Glasgow Coma Scale (GCS) > 8 [15]. 

### 2.2. Secondary Injury

In a healthy person, cerebral blood flow (CBF) is autoregulated, and is directly proportional to cerebral perfusion pressure (CPP) and inversely proportional to cerebral vascular resistance (CVR). After head trauma, the CVR often increases, and autoregulation mechanisms can be impaired. With autoregulation disrupted, there can be a subsequent increase in arterial pressure, leading to increasing CBF, capillary hydrostatic pressure, worsening cerebral edema, and increased intracranial pressure (ICP) [16]. 

Traumatic brain edema can be classified as vasogenic, cytotoxic, and osmotic. Vasogenic edema occurs due to a disruption in the blood–brain barrier (BBB) that causes increased permeability of the BBB. The mechanical injury and auto-destructive mediators that result from TBI disrupt the tight junctions of cerebral endothelial cells, which normally control passage of proteins into the brain [17]. Cytotoxic brain edema involves the accumulation of intracellular fluid within astrocytes and neurons. During physiologic conditions, the inward flow of ions is balanced by energy-dependent pumps like Na+/K+ ATPase. After TBI, there is neuronal activation and cellular lysis increasing Na+ influx, for which the Na^+^/K^+^ ATPase becomes unable to compensate. The combination of increased adenosine triphosphate (ATP) demand and surrounding cerebral ischemia leads to Na+/K+ ATPase failure and continued uptake of osmotically active ions. Osmotic brain edema occurs due to rapid changes in osmolarity within the brain or interstitium that cannot be compensated for. Serum hyposmolarity, associated with the syndrome of inappropriate antidiuretic hormone secretion and the serum hyperosmolarity associated with cerebral ischemia, causes osmotic cerebral edema [16,17]. Astrocytes and microglia play a key role in neuroinflammation following TBI. These cells secrete several cytokines, chemokines, and growth factors, which lead to microenvironmental changes in areas of trauma, affecting cellular damage and repair [18]. 

Even if autoregulation is preserved, in trauma patients, the autoregulation curve shifts to the right, meaning a higher CPP is required to maintain adequate CBF, potentially worsening ischemia [19]. Tissue hypoxia after TBI can be attributed to microvascular ischemia in addition to areas of direct mechanical injury [20]. In areas of temporary ischemia, reperfusion can contribute to additional secondary injury. Reperfusion to ischemic areas increases the presence of reactive oxygen species like superoxide (O_2_^−^), which can react with the neurotransmitter nitric oxide (NO). This forms peroxynitrite (ONOO-), which leads to intracellular damage on the level of deoxyribonucleic acid (DNA), proteins, and lipids, contributing to possible cell death [14,17]. 

A proposed mechanism for microvascular ischemia supported by pathologic studies of brain samples is intravascular thrombosis. Tissue factors released by the injured brain can initiate an extrinsic coagulation cascade as damaged cerebral vessels release microthrombi. Consumption of coagulation factors in this process can contribute to the coagulopathy associated with TBI [21]. Coagulopathy contributes to the conversion of initial contusions into hemorrhagic lesions, which was found in approximately half of TBI patients who received serial CT scans [13]. This progression generally occurs within the first 12 h of injury but can occur up to 3–4 days later. Larger contusions where patients initially present with a lower GCS score are more likely to progress into hemorrhagic lesions and require surgical decompression [13]. 

An additional long-term consequence of TBI is the potentially permanent damage to executive function and the development of depression/anxiety [5]. A prospective study of 1084 TBI patients showed that, at 12 months postinjury, 22% of patients reported a new psychiatric disorder, with depression (9%) and generalized anxiety (9%) being the most common [22]. Although these mechanisms are unknown, it has been hypothesized to be linked to the neurotransmitter abnormalities that occur after TBI [23]. In acute TBI, there is an increase in extracellular glutamate, a decrease in glutamate uptake by cells, and downregulation of N-methyl-D-aspartate (NMDA) receptors within hours. Chronically, there is a sustained depression of glutamate. Acute injury causes a shift in synaptic vs. extra-synaptic gamma–aminobutyric acid (GABA) receptors. Over time, in the cortex, altered GABA receptors can change the structure of pyramidal neurons, and changes to GABA receptors can be epileptogenic in subcortical areas [24]. After initial trauma, there is an unregulated release of acetylcholine but there is a subsequent decrease in transporter density and receptor binding which can happen within one hour of injury. Norepinephrine turnover is initially accelerated, but the subsequent downregulation of receptors and decreased signaling can last for weeks after the initial injury. Dopamine levels increase acutely after TBI but dopamine release decreases chronically along with a decrease in dopamine recycling. In addition, serotonin receptors and transporters were both decreased after TBI [25]. The role of viral reactivation in secondary injury following TBI is unclear, However, it may affect neuroinflammatory pathways leading to microenvironmental modulation [26]. A study of 344 patients with severe TBI found that a specific monocyte signature with decreased antiviral response signaling was associated with increased herpes simplex virus reactivation and poor clinical recovery at 6 months [27].

## 3. Clinical Management

### 3.1. Initial Assessment

The Brain Trauma Foundation provides evidence-based guidelines for the management of TBI. The main focus of clinical management is the prevention and prompt management of intracranial hypertension and secondary brain injury [28]. Some common symptoms associated with TBI include headache, nausea, vomiting, dizziness, drowsiness, mental status changes, vision or speech abnormalities, balance issues, etc. [29]. A non-contrast-computed tomography (CT) of the head is the standard-of-care initial imaging study for moderate–severe TBI. CTs have high sensitivity and specificity for intracranial hemorrhage, edema, midline shift herniation, and skull fractures. They are quick studies that are readily available, and they lack major contraindications. In specific populations such as children, the exposure to radiation must be considered prior to deciding to obtain a head CT [30]. A magnetic resonance imaging (MRI) study is indicated in situations where a non-contrast head CT is unable to provide images that correlate to neurological deficits. MRI is more sensitive at demonstrating diffuse axonal injury, grading intracranial hemorrhage, detecting contusions, microhemorrhages, and brainstem injuries [4,31]. 

### 3.2. Monitoring

The current recommendations regarding ICP monitoring are based on moderate–low-quality/weak evidence. ICP monitoring is recommended in patients with severe TBI who undergo surgical evacuation postoperatively if the patient meets any of the following conditions: preoperative GCS motor response score of 5 or less, preoperative hemodynamic instability, concerning signs on preoperative CT imaging (compressed basal cisterns, midline shift greater than 5 mm), preoperative anisocoria or bilateral mydriasis, intraoperative cerebral edema, or new lesions on postoperative CT imaging [32]. For nonoperative candidates, ICP monitors are recommended when the CT scan demonstrates signs of increased ICP, if serial neurological examination is not possible, and during extracranial surgery. ICP monitoring devices are not recommended in patients with a normal CT scan and no evidence of neurologic deficits, as risks such as infection, intracranial hematoma, and failed catheter placement outweigh the benefits in those patients at low risk for increased ICP [33].

Given the associated poorer outcomes and increased mortality with intracranial hypertension, it is reasonable to measure ICP [34]. Current guidelines recommend ICP monitoring in TBI patients with GCS 3–8 after resuscitation and an abnormal CT scan [28]. The current gold standard for ICP monitoring is the intraventricular catheter, which measures global ICP, assuming there is no CSF flow obstruction. Epidural, subdural, and subarachnoid catheters are rarely used, since they are less accurate than the intraventricular catheter [35]. Noninvasive methods to measure ICP include brain CT, brain MRI, transcranial Doppler ultrasonography, optic nerve sheath diameter, and tympanic membrane displacement [35,36]. Brain CT offers the fastest and most cost-effective method, while a brain MRI offers a more accurate assessment of soft tissue lesions [36]. Transcranial Doppler can be performed at the patient’s bedside but is highly operator-dependent [37]. Optic nerve sheath diameter ultrasonography is cheap and not time consuming but has a high degree of observer variability [38], while the tympanic membrane displacement is largely inaccurate and unreliable [39]. Thus, the intraventricular catheter remains the gold standard for measuring ICP [36].

An external ventricular drain (EVD) is a device placed into the lateral ventricle that can provide ICP monitoring and allows for the drainage of CSF, either continuously or intermittently, to directly decrease ICP [40]. Based on low-quality indirect evidence, the current guidelines recommend the continuous drainage of CSF with an EVD zeroed at the midbrain to lower ICP compared to intermittent drainage, and CSF drainage to lower ICP in patients with an initial GCS < 6 during the first 12 h following TBI [28,41]. Intraparenchymal ICP monitors are placed directly into the brain tissue, but they may not accurately measure pressure in the CSF due to intracranial pressure gradients that occur following TBI. Currently, there is no proven benefit to ICP monitoring but current recommendations based on low-quality evidence suggest ICP monitoring in the management of severe TBI to reduce 2-week mortality [40,41]. Another strategy for monitoring ICP monitoring is the arteriovenous oxygen content difference (AVDO2) between a central line (typically placed in the jugular bulb) and an arterial line. This newer method of ICP monitoring currently lacks solid evidence but there is low-quality indirect evidence that 6-month outcomes are more favorable and lower mortality has been recorded when AVDO2 is maintained above 50% [28].

Once the ICP is determined, a staircase therapeutic approach may be used to escalate interventions as necessary for each individual patient [42]. Guidelines recommend maintenance of ICP below 22 mmHg and a target CPP between 60 and 70 mmHg [28,36]. For blood pressure monitoring, hypotension and hypertension should be quickly recognized. Guidelines recommend a systolic blood pressure (SBP) at ≥100 mm Hg for patients 50–69 years old, and at ≥110 mm Hg or above for patients 15–49 or >70 years old; this may be considered in order to decrease mortality and improve outcomes [28,43]. A patient may be considered hypotensive if the SBP is less than 100–110 mmHg [44]. If a patient’s MAP is >110 mmHg, SBP > 150 mmHg, or ICP > 20 mmHg, then SBP should be gradually lowered so as to not decrease CPP significantly [43]. Short-acting titratable agents, such as nicardipine or labetalol, may be used [45]. 

Recent studies investigating the feasibility of biomarkers in blood such as S100B and GFAP for the assessment of TBI have not been demonstrated to have beneficial clinical applications [46]. Measurement of S100B levels in CSF of patients with severe TBI showed that higher S100b levels were associated with higher acute mortality, worse Glasgow Outcome Scale (GOS), and lower disability rating scale 6 months after injury [47]. Due to low specificity, S100B was not included in guidelines regarding TBI management [46]. GFAP levels in patients with severe TBI trended over 5 days postinjury, compared to healthy controls, showed a gradual reduction for the first 3 days of admission until the day 4, when the protocol dictated discontinuation of hypothermia and initiation of rewarming. Significantly higher GFAP levels were correlated with death and worse outcomes [48]. Comparison of six serum biomarkers (S100B, NSE, GFAP, UCH-L1, NFL, and t-tau) within 24 h of TBI found that GFAP had the highest discrimination for predicting CT abnormalities with a 99% likelihood of better discrimination. However, in mild TBI, the discrimination increased from 0.84 to 0.89; although this was statistically significant, it is only a small increase, with unknown clinical significance [49].

### 3.3. Surgical Management 

After general measures have been applied, surgical resection of mass lesions and CSF drainage should be considered [50]. According to the 2017 guidelines for the management of severe TBI, initial surgical evacuation of an epidural hematoma (EDH) is recommended if the patient has a GCS less than 9, clot thickness greater than 15 mm, midline shift greater than 5 mm, or focal neurological deficits [28]. Additionally, in patients with subdural hematoma (SBH), surgical evacuation is considered if the SDH is larger than 1 cm, there is midline shift 5 mm or larger, a GCS less than 8, an ICP greater than 20 mm Hg, or rapid neurologic decline. If patients are surgical candidates, studies have shown a significant mortality benefit if surgical evacuation is performed within 4 h of injury [41]. 

In patients with refractory elevated ICP, a decompressive craniectomy (DC), where a large portion of the skull is temporarily removed, is considered, but it is a controversial practice. The Decompressive Craniectomy in Diffuse Traumatic Brain Injury (DECRA) trial investigated secondary DC as a treatment for early-refractory ICP elevation [51]; the Randomized Evaluation of Surgery with Craniectomy for Uncontrolled Elevation of Intracranial Pressure (RESCUEicp) trial investigated secondary DC as a treatment for late-refractory ICP elevation. Both DECRA and RESCUEicp demonstrated reduction in both ICP and duration of intensive care with DC [52]. However, both DECRA and RESCUEicp demonstrated unfavorable long-term outcomes in terms of neurological function, as measured by the Extended GOS. As a result, the current recommendations based on moderate evidence are to perform secondary DC for late-refractory ICP elevation but not early-refractory ICP elevation [51,52,53]. 

### 3.4. Ventilation

Mechanical ventilation is frequently utilized in patients with TBI admitted to the intensive care unit (ICU) due to respiratory failure, the inability to protect the airway due to the loss of the airway-protective reflexes, decreased respiratory drive, and increased risk of pulmonary complications such as pneumonia or acute respiratory failure [54,55,56,57]. There is no evidence to support the need for mechanical intubation in patients with TBI specifically; thus, clinical practice is commonly based on parameters such as level of consciousness, severe agitation and combativeness, loss of airway-protective reflexes, GCS ≤ 8, clinical evidence of brain herniation, and significant ICP elevation, among other factors [28]. During induction and intubation, the patient should be adequately sedated to eliminate reflexes that may increase ICP, such as coughing or vomiting [58]. Once intubated, the head of the bed should be elevated to between 30° and 45°, and the patient’s head should be positioned at the midline to prevent the compression of the internal jugular vein [59].

Guidelines recommend hyperventilation as a temporary measure to reduce elevated ICP but it should be avoided within the first 24 h following the trauma as CBF is often reduced. Hyperventilation with partial pressure of arterial carbon dioxide (PaCO2) of ≤25 mm Hg is not recommended [28]. The use of lower tidal volumes (6–7 mL/kg) in mechanical ventilation has been associated with a lower risk of ventilator-induced lung injury as compared to higher tidal volumes (median of 9 mL/kg) and a reduced risk of acute respiratory distress syndrome [60,61]. Studies have shown a positive correlation between the use of high PEEP > 5 cm H_2_O and increased ICP in patients with TBI, as high PEEP can lead to increased positive intrathoracic pressure and impair venous return, leading to increased ICP and reduced CPP [62,63]. However, high PEEP has been shown to be beneficial in patients with acute respiratory distress syndrome and has been applied in patients with neurologic injuries without a significant effect on ICP or CPP [64]. 

Several studies have tried to develop specific criteria for the extubation of patients with TBI based on the level of arousal and the use of GCS ≥ 8; this is because delayed extubation is associated with multiple complications, such as increased risk of ventilator-associated pneumonia, increased length of stay in ICU, and increased cost [65,66]. Age < 40, visual pursuit, ability to swallow, and a GCS > 10 on the day of extubation were independent markers of successful extubation [67]. Tracheostomy has been advocated as a valid option for patients with TBI giving the complicated nature of the extubation process in this patient population. However, the benefit of early vs. late tracheostomy remains debatable [68]. In addition, guidelines report that early tracheostomy is recommended to reduce the overall duration of mechanical ventilation but does not reduce the incidence of nosocomial pneumonia [28]. A systematic review and meta-analysis reported that early tracheostomy in TBI patients may lead to a reduction in mechanical ventilation duration, ventilator-associated pneumonia, ICU admission, and hospital length of stay, but may lead to an increased risk of mortality [69].

### 3.5. Fluid Therapy

Patients with TBI and increased ICP can benefit from therapy with hyperosmolar agents such as mannitol and hypertonic saline [36,70]. Mannitol has shown to have a significant benefit in reducing ICP in patients with TBI when given in a dose-dependent manner. It also improves CBF and CPP as well as reducing the inflammatory response when given to patients with TBI [70]. The standard ICP-lowering dose of Mannitol is 0.25–1 g/kg every 6 h [71]. Hypertonic saline is less capable of crossing the BBB, so it reduces ICP by reducing cerebral edema in addition to increasing MAP and subsequently CPP [72]. However, no difference between mannitol and hypertonic saline has been observed regarding 6-month mortality. The status of mannitol as the “gold standard” has thus come into question [36]. A pooled report of seven studies showed hypertonic saline was more effective in reducing mean ICP change after TBI at 60 and 120 min after intervention with no changes in serum osmolarity [73]. Guidelines recommend avoiding mannitol administration prior to ICP monitoring in patients with signs of transtentorial herniation or progressive neurologic deterioration that are not attributable to extracranial causes [28]. 

After the failure of other measures, barbiturates may be used to decrease ICP, decrease the cerebral metabolic rate of oxygen, and improve the overall oxygenation of cerebral tissues [74]. High-dose barbiturate therapy is only recommended for controlling increased ICP refractory over standard medical and surgical treatment as long as hemodynamic stability is maintained [36]. The use of barbiturates to induce burst suppression measured by EEG is not recommended for preventing the development of elevated ICP [28]. The administration of propofol is recommended for ICP control but does not improve mortality or 6-month outcomes, and high-dose propofol can lead to increased morbidity [28]. 

### 3.6. Anti-Thrombolytics 

Patients with TBI who were found to have higher levels of fibrinogen degradation products (i.e., D-Dimer) upon admission were associated with a higher risk of progressive hemorrhagic injury [75]. The Clinical Randomization of an Antifibrinolytic in Significant Hemorrhage-2 (CRASH-2) trial supported the early administration (within 3 h) of tranexamic acid (TXA) in patients with TBI to reduce the likelihood of death due to extracranial bleeding. Patients were allocated to TXA (n = 10,096) and placebo (n = 10,115) groups and all-cause mortality at 28 days was lower in the TXA group (1463 patients (14.5%) in the TXA group vs. 1613 patients (16.0%) in the placebo group; relative risk (RR) 0.91; 95% confidence interval 0.85 to 0.97; *p* = 0.0035). When data were further stratified, there was a statistically significant reduction in death in groups receiving TXA < 1 h from injury and within 1–3 h from injury, but not those who received TXA > 3 h from injury [76]. 

The Clinical Randomization of an Antifibrinolytic in Significant Head Injury-3 (CRASH-3) trial hypothesized that the administration of TXA within 3 h of injury could reduce death and disability due to intracranial bleeding. TBI patients (12,737 total) were randomly assigned to receive TXA or placebo within 3 h of admission and the two groups were stratified by severity of brain injury by Glasgow Coma Score and pupil reactivity. There was a statistically significant reduction in head-injury-related death in the TXA group versus placebo in patients with mild–moderate head injury [RR 0.78 (95% CI 0.64–0.95)] but not in patients with severe head injury [0.99 (95% CI 0.91–1.07)]. In patients with mild–moderate head injury, earlier treatment was more effective than later treatment (*p* = 0.005) but the same was not noted in severe head injuries (*p* = 0.73). The risk of blood clots [RR 0.98 (0.74–1.28)] and seizures [1.09 (95% CI 0.90–1.33)] were similar in both groups [77].

### 3.7. Temperature Control 

Fever occurs in approximately 40–70% of TBI patients [78,79,80]. These fevers may be related to several causes, including the endogenous release of pyrogens from damaged neurons, disrupting the hypothalamic set point [78,81]. Bao L et al. defined fever burden as a patient’s highest axillary temperature minus 37 °C, and reported that fever burden was negatively associated with GOS and an independent prognostic factor for TBI [82]. Fever burden increased as GOS scores decreased from 5 to 2, whereas a GOS score of 1 showed a significantly lower fever burden. Patients in higher GOS groups were younger, had more normal pupil reactivity, and lower median fever burden. Fever burden was identified as an independent predictor of poor outcomes after TBI (OR 1.098; 95% CI: 1.031–1.169; *p* = 0.003) [82].

Treating fever with induced normothermia may lessen cerebral damage in patients with severe brain injury [83]. The application of induced normothermia is complicated by shivering, a physiologic thermoregulatory response [84]. Shivering thus may limit the effectiveness of the therapeutic cooling meant to attenuate cerebral damage [84,85]. Oddo et al. measured the regional brain tissue oxygen tension (PbtO_2_) in 15 patients with brain injury while inducing normothermia [85]. Lower PbtO_2_ levels are associated with poorer outcomes after severe brain injury. Using water-circulating gel-coated pads, the patient’s water temperature was adjusted based on the difference between the patient’s rectal temperature and the goal temperature. Their observational study found that PbtO_2_ decreased from baseline during shivering episodes with a baseline of 34.1 ± 7.3 mmHg and 24.4 ± 5.5 mmHg during shivering episodes (*p* < 0.001) [85]. This may suggest that shivering during induced normothermia is associated with a significant reduction in PbtO_2_ and that the associated decrease of pO2 correlates with the level of cooling [85]. The authors acknowledge that a causal relationship cannot be drawn, but they argue that shivering during induced normothermia may impair pO2 in those with severe brain injury. Thus, there may be a benefit to reducing shivering via therapeutic interventions like buspirone, meperidine, or neuromuscular-blocking agents, as well as the use of a computerized thermoregulatory system to more accurately adjust a patient’s temperature during induced hypothermia [85,86]. 

### 3.8. Nutritional Support and Glucose Control 

Patients with severe brain injury enter an acute catabolic state and require nutrition to preserve skeletal muscle mass, vital organ function, and cerebral metabolic homeostasis [87]. Ideally, nutrition therapy should be started within the first 24 h of injury [88]. Wang et al. performed a systematic review of 13 randomized controlled trials which showed the benefits of early nutrition for mortality, functional outcomes, and infection. Pooled data showed a significant reduction in mortality with early feeding (relative risk [RR] = 0.35; 95% CI, 0.24–0.50), functional outcomes (RR = 0.70; 95% CI, 0.54–0.91), and infectious complications (RR = 0.77; 95% CI, 0.59–0.99) [89]. Parenteral nutrition has classically been associated with higher risk of infection, immunosuppression, hyperglycemia, and hepatic steatosis [90]. Enteral nutrition can provide a better quality of micro- and macronutrients. Kurtz et al. advocate for the use of early enteral nutrition [88]. Clinical guidelines recommend nutrition aiming to attain basal caloric replacement by the fifth day, or at most the seventh day, postinjury to decrease mortality [28]. In ventilated patients, transgastric jejunal nutrition is preferred to reduce the incidence of ventilator-associated pneumonia. 

When TBI patients are receiving mannitol or hypertonic saline, clinicians also need to monitor fluid and electrolyte derangements [28,36]. Nutritional therapies should avoid excessive fluid, electrolyte, and glucose shifts [88]. Both hypoglycemia and hyperglycemia should be avoided in patients with TBI. Guidelines for critically ill patients suggest that a blood glucose < 100 mg/dL should be avoided. Blood glucose >180 mg/dL should also be avoided, and insulin therapy should be initiated in patients with neurological injury such as TBI, ischemic stroke, intraparenchymal hemorrhage, or SAH [28,36].

Tight glucose control is commonly advocated for in the treatment of neurocritical patients. The American Heart Association/American Stroke Association recommend serum glucose concentrations in the range of 140–180 mg/dL for patients with acute stroke [91]. Hyperglycemia following TBI is correlated to injury severity and an independent predictor of mortality [92,93,94]. TBI causes elevated serum glucose levels, and approximately 87% of TBI patients entering the ICU have hyperglycemia [95]. Although hyperglycemia is deleterious in TBI patients, hypoglycemia in patients with severe TBI has been shown to correlate to worse functional outcomes 6 months after the trauma [96]. Therefore, studies do not support the aggressive treatment of hyperglycemia [97]. The Normoglycemia in Intensive Care Evaluation and Surviving Using Glucose Algorithm Regulation (NICE-SUGAR) multicenter international trial randomized 6104 medical and surgical ICU patients to either moderate or intensive glucose control. They defined intensive glucose control as a target blood glucose range of 81–108 mg/dL and moderate glucose control as being <180 mg/dL. Of those patients, 391 satisfied the diagnostic criteria and were followed [98]. At 2 years, 58.7% of the patients in the intensive group and 53% of the patients in the moderate group had favorable neurological outcomes (OR—1.26; 95% CI, 0.81–1.97; *p* = 0.3). These results from the NICE-SUGAR multicenter study suggested no benefit of intensive vs. moderate glucose therapy in the ICU [98]. To further corroborate these results, a systematic review and meta-analysis of 10 randomized controlled trials analyzed the data of 1066 TBI patients [97]. Similarly, patients were divided into intensive and conventional glucose control, defined as 79.2–120.6 mg/dL and 151.2–216 mg/dL, respectively. The results showed no association with ICU mortality (RR = 0.93; 95% CI, 0.68–1.27; *p* = 0.64) or hospital mortality (RR = 1.07; 95% CI, 0.84–1.36; *p* = 0.62). However, the conventional group did show a higher risk of poor neurological outcomes, which the authors defined as a GOS of 1–3, measured at various times from 3–24 months (RR = 1.1; 95% CI, 1.001–1.24; *p* = 0.047) [97]. According to the American Diabetes Association, blood glucose levels should be maintained between 80 and 130 mg/dL in critically ill patients, including TBI patients [99]. However, TBI patients may incur worse outcomes if blood glucose level is maintained >150 mg/dL [100,101]. Therefore, the exact range for serum glucose levels in TBI patients has not been clearly determined, but it seems evident from the literature that serum glucose levels below 80 mg/dL and above 180 mg/dL are deleterious.

### 3.9. Seizure Prophylaxis

The literature has described seizures after TBI for years, but its incidence has not been definitively determined. Seizures worsen the functional outcome following TBI [102]. TBI is considered one of the factors in etiologies of up to 20% of symptomatic epilepsy cases [103]. Posttraumatic seizures (PTS) and posttraumatic epilepsy (PTE) are terms used to describe seizures following TBI. PTS refers to seizures that occur one week following TBI, whereas PTE refers to unprovoked seizures occurring at least one week after TBI. Some studies refer to seizures occurring after TBI as PTS for up to four weeks following head trauma [104]. The incidence of PTE was 2% for mild brain injury, 4% for moderate brain injury, and 15% for severe brain injury [105]. PTE may further exacerbate deficits in memory and cognition, development of PTSD or depression, and damage to the cerebrovascular system or BBB [106]. The seizures associated with PTE can also make treating the primary injury more difficult and increase the costs associated with treatment [105]. Thus, the prevention of PTE is an important clinical goal. 

While the treatment for PTE in patients with TBI has been controversial, phenytoin has historically been the therapy of choice. Huo et al. performed a meta-analysis to determine the best treatment for PTE in patients with TBI [107]. This meta-analysis of 25 studies and a total sample size of 6466 compared the direct and indirect outcomes of several treatment regimens (phenytoin and phenobarbital, levetiracetam, phenytoin, phenytoin and levetiracetam, lacosamide, and valproic acid) [107]. Their findings showed that phenytoin could reduce the incidence of early seizures, but not the incidence of late seizures or mortality [108]. When comparing effectiveness, these various therapies led to a reduced incidence of early PTE compared with placebo [109]. Valproic acid was not shown to reduce early PTE [110]. Their analysis showed no significant difference between phenytoin and levetiracetam in the prevention of early and late seizures, treatment-related adverse events, and mortality in TBI patients [111]. In the studies evaluating treatment-related adverse outcomes, levetiracetam and phenytoin had more adverse effects compared to the placebo. In a systematic evaluation by Xu et al., levetiracetam was found to have a better safety profile than phenytoin [109]. They also found that levetiracetam reduced the average length of ICU stays when compared to phenytoin, but the overall hospital length of stay was the same for levetiracetam and phenytoin [109]. However, there is currently a lack of sufficient evidence to support levetiracetam’s efficacy compared to phenytoin in the prevention of PTS [28]. Thus, clinical guidelines recommend phenytoin to decrease the incidence of PTS when the benefits outweigh the risks of the treatment. However, the use of phenytoin or valproic acid is not recommended for the prevention of PTE [28]. 

### 3.10. Steroids

Steroids appear to inhibit the gene expression of proinflammatory molecules [112]. Steroids are not recommended in the treatment of severe TBI, and high-dose methylprednisolone led to increased mortality [28]. To ascertain the benefits of steroid use following head injury, the MRC CRASH trial randomly assigned 10,008 adults to either a treatment comprising 2 g of methylprednisolone over 1 h followed by 0.4 g methylprednisolone for the next 48 h or to a placebo [113]. At 2 weeks, the risk of death was higher in the group given corticosteroids (1052 [21.1%] vs. 893 [17.9%] deaths; relative risk 1.18 [95% CI 1.09–1.27]; *p* = 0.0001) [113]. The risk of death within 6 months with corticosteroid use was 25.7% vs. 22.3% with placebo (95% CI 1.07–1.24; *p* = 0.0001) [114]. Similar findings were present when comparing the corticosteroid group vs. placebo for risk of death or severe injury with 38.1% vs. 36.3%, respectively (95% CI 0.99–1.10; *p* = 0.079) [114]. When comparing subgroups, the effect of corticosteroids was not affected by the severity of the injury or the time of the injury [113]. 

### 3.11. Infection and Deep-Vein Thrombosis Prophylaxis

Clinical guidelines recommend the use of antimicrobial-impregnated catheters to prevent catheter-related infections during EVD drainage, but do not support the use of povidone–iodine oral care to reduce ventilator-associated pneumonia or early tracheostomy to decrease nosocomial pneumonia risk [28]. In addition to mechanical deep-vein thrombosis prophylaxis, pharmacologic prophylaxis may be indicated if the brain injury is stable, and the benefits outweigh the increased risk of intracranial hemorrhage. Low-molecular-weight heparin or low-dose unfractionated heparin in combination with mechanical deep-vein thrombosis prophylaxis is recommended. Studies have not reported sufficient evidence to support a certain pharmacologic agent, dose, or schedule for pharmacologic prophylaxis of deep-vein thrombosis in TBI patients [28].

## 4. Emerging Treatments 

Several research studies have provided insight into potential novel treatments that may improve outcomes in patients with TBI such as anti-excitotoxic agents, calcium channel blockers, nitric oxide, statins, S100B protein, erythropoietin, endogenous neuroprotectors, anti-inflammatory agents, and stem cell and neuronal restoration agents, among others.

### 4.1. Anti-Excitotoxic Therapy

A major form of tissue damage, manifested by deleterious subclinical seizures, usually occurs early in severe TBI. This process is attributed to excitotoxicity, and NMDA receptors have been targets for anti-excitotoxic therapy [115]. NMDA receptor antagonists, such as amantadine and memantine, were tested in animals with TBI and found to block extrasynaptic NMDA receptors while sparing synaptic NMDA receptor function [116,117]. NMDA receptor antagonists could protect neurons against glutamate excitotoxicity in the acute phase of TBI [118]. Amantadine (100–400 mg/d dose) may improve cognitive function and increase arousal when given within 12 weeks after the TBI [118,119].

### 4.2. Calcium Channel Blockers

Increased levels of intracellular calcium play a critical role in the cascade of cell death following TBI [70,120]. The L-type and N-type of calcium channel blockers have shown promising benefits in preventing the cellular damage caused by TBI through their mechanism of neutralizing the intracellular calcium level [120]. Nimodipine is an L-type calcium channel blocker known for its well-established neuro protective mechanism [121]. An N-type calcium channel blocker, ziconotide (SNX111), has been shown to improve mitochondrial function if administered during the period of 15 min–6 h following TBI [70]. 

### 4.3. Nitric Oxide

TBI causes vascular dysregulation, leading to an increased vulnerability to a secondary injury as CBF is usually reduced after severe TBI [122,123]. Nitric oxide is a potent vasodilator and studies have shown improved outcome in patients with TBI treated with nitric oxide, which led to improved blood flow through the collateral circulation [119]. However, caution is essential when administering nitric oxide as it may cause severe hypotension, leading to reduced CBF.

### 4.4. Statins

Statins are β-Hydroxy β-methylglutaryl-CoA (HMG-CoA) reductase inhibitors which have an anti-inflammatory effect and upregulate the nitric oxide synthesis pathway. Thus, statins lead to increased nitric oxide production, improving capillary patency [124]. Studies with statins have suggested an improved outcome for patients with TBI but its role in regulating CBF is unclear [125].

### 4.5. S100B Protein

Produced by glial cells, S100B is a calcium-binding protein that was detected in serum following damage of the BBB in patients with TBI. It has dose-dependent effects on neurons, acting as a neuroprotective agent at small doses and increasing inflammation and worsening neuronal survival at high doses [70].

### 4.6. Erythropoietin

Erythropoientin (EPO) is a glycoprotein that plays a major role in the regulations of erythropoiesis in the hematopoietic system [126]. EPO also plays a role in the central nervous system as the expression of both EPO and its receptor are widespread in the brain [127,128]. EPO appears to promote neuroprotection through binding to EPO receptors and activating the JAK-2/NF-kB and PI3K signaling pathways [70,129]. JAK-2 phosphorylation activates the PI3K/AKT and Ras/MAPK pathways and promotes STAT-5 homodimerization, which has been shown to have antiapoptotic and neurotrophic effects [130,131]. Patients with TBI have been found to have increased expression of EPO receptors in the neurons, glia, and endothelial cells [70]. Exogenous EPO has been found in the brain parenchyma despite the fact that its molecular weight is larger than that of the BBB, suggesting that it plays a role in neuroprotection after TBI [132]. EPO shows anti-excitotoxic, antioxidant, antiedematous, and anti-inflammatory effects in TBI, and a reduced number of progenitor cells has been linked to reduced or lacking EPO receptors [133,134].

### 4.7. Adenosine

Adenosine is produced by several pathways by damaged tissues in patients with TBI, likely from the breakdown of the mRNA via 2′,3′cAMP pathway, which plays a major role after TBI [135]. Adenosine acts on different receptors (A1, A2A, A2B, and A3) and engages signal transduction that is neuroprotective in TBI [136]. Activation of A1 receptors has been found to decrease posttraumatic seizures [137]. An increase in the level of adenosine and/or upregulation of A1 receptors by the chronic and prophylactic use of short-acting A1 receptor blockers (e.g., caffeine) has been associated with improved outcomes in TBI patients with caffeine in their CSF at the time of injury [138].

### 4.8. Therapies to Improve Mitochondrial Function

Mitochondria have multiple vital functions such as the generation of ATP and production of reactive oxygen species, which are important in the regulation of life and death decisions in cells [139]. Mitochondrial dysfunction is associated with adverse outcomes in patients with TBI via inflammatory and vasoactive mediators [139]. 1-methyl-4-phenyl-1,2,3,6-tetrahydropyridine (MPTP) inhibitors have been the focus of some studies, and cyclosporine, an MPTP inhibitor, has been given safely to patients with severe TBI [140,141].

### 4.9. Anti-Inflammatory Therapy

Secondary injuries in patients with TBI are mediated by an inflammatory response mediated by cytokines, chemokines, microglial activation, and recruitment of circulating leukocytes [142,143]. Tumor necrosis factor (TNF) alpha synthesis inhibitors (thalidomide analogs) and etanercept (fusion protein that binds and inhibits TNF alpha) have shown benefits early in mild TBI in preclinical studies [144,145].

### 4.10. Neuronal and Neurovascular Regeneration

Neuronal and neurovascular regeneration is thought to play a potential role in brain recovery after TBI. Injection of Thymosin B4 (a peptide with G actin-sequestering action) in animal models has been found to enhance the proliferation of neuronal cells, promoting angiogenesis and neuronal cell differentiation [146]. The process of neuronal proliferation and differentiation has a peak at 2–5 days after TBI, while some studies have extended this timeframe to 14 days [147]. However, neurogenesis in the adult brain has limited capacity and appears to occur in just a few areas of the brain [148,149].

### 4.11. Neurorestorative Therapy

Stem cell-based therapy has shown beneficial effects in improving recovery in patients with neurological injuries including those with TBI [150,151]. Stem cells exert their benefits by enhancing neurogenesis, angiogenesis, and immunoregulation by secreting chemokine and growth factors [152,153]. Mesenchymal stem cell transplantation’s therapeutic effects in TBI recovery have been demonstrated in animal models [153,154,155]. Remarkable improvements in the neurological function of patients with a sequelae of TBIs after umbilical cord mesenchymal stem cell transplantation has been reported in a recent study [155]. Furthermore, emerging evidence has suggested that neurorestoration is most likely the mechanism underlying mesenchymal-stem-cell-transplantation-induced TBI recovery rather than neuroreplacement. This process likely occurs by releasing growth factors such as fibroblast growth factor 2, vascular endothelial growth factor (VEGF), and brain-derived neurotrophic factor (BDNF). These growth factors enhance neurogenesis, angiogenesis, and synaptogenesis [156].

Exosome therapy from neural stem cells are viewed as a promising alternative to stem cells as they readily traverse the BBB, are more stable, are immune tolerant, and have a lower tumorigenic risk [157]. Neural stem cell exosomes upregulate VEGF, which promotes angiogenesis and may contribute to neurorestoration [158]. In rats who suffered TBI, exosomes derived from human neural stem cells improved motor function recovery by inhibiting reactive astrocytes and increasing the expression of the doublecortin protein, a microtubule protein involved in regulating neuroinflammation and promoting neurogenesis [159]. Multiple in vitro studies have demonstrated that neural stem cell exosomes inhibit neural apoptosis [160,161,162].

### 4.12. Combination Therapy

TBI is a multifactorial and complex pathophysiologic entity that will likely require combination therapy to address all the secondary injuries resulting from it [163]. New recommendations suggest the combination of therapies with complementary targets and effects rather than focusing on a single target with multiple therapies [164]. One of the promising combination therapies is a combination of the anti-inflammatory agent minocycline and the glutathione precursor N-acetyl cysteine (NAC). Promising outcomes in some TBI models and in blast-induced TBI in humans have been reported with this combination therapy [165,166,167]. Current studies are underway to prove the benefit of another combination therapy of the organic acid transporter with a multidrug-resistance-associated protein inhibitor, probenecid, and NAC, which aim to improve the NAC bioavailability and antioxidant reserves in patients with TBI by overcoming the BBB. Preclinical data show that probenecid increases NAC brain penetration in juvenile rats by 2–3 times, as early as one hour after injury [168].

### 4.13. Enriched Environment

Providing adequate rehabilitation to patients with TBI has played a significant role in helping these patients attain a more functional lifestyle. Individuals with TBI are known to have a high incidence of depression; thus, the provision of an enriched environment plays a critical role in rehabilitative efforts. Numerous animal studies have shown significant improvement both in neuroanatomy and neurobehavior by simply providing these animals with large cages and allowing them an opportunity for social interactions, sensory stimulation, and exploratory behavior [169]. Thus, applying an enriched environment for individuals recovering from TBI has positive implications in neurorehabilitation.

## 5. Conclusions

TBI is a major cause of death and disability worldwide, accounting for 27.08 million cases in 2016 with an estimated cost of USD 76.56 billion in the United States in 2010. Thus, the treatment of TBI is paramount in reducing morbidity and mortality among populations worldwide. The clinical management of TBI focuses on treating the primary injury and stabilizing the patient, with the goal of preventing secondary injury, such as cerebral ischemia, and further neuronal injury. Continuous ICP monitoring guides clinical management in severe TBI, and surgical intervention such as DC and CSF drainage should be considered in the early management of TBI.

Clinical guidelines offer specific parameters for hemodynamic management, ventilation, fluid therapy, seizure prophylaxis, nutritional support, temperature control, etc. Although there have been significant advancements in the management of TBI, there remain numerous areas in need of further research, such as the cytotoxic and inflammatory pathways contributing to secondary injury following head trauma. Several trials have shown promising evidence in the use of adjunctive therapy in improving outcomes and functional status following TBI. Emerging research studies have provided insight into potential novel treatments that may improve outcomes for patients with TBI, such as anti-excitotoxic agents, calcium channel blockers, nitric oxide, statins, S100B protein, erythropoietin, endogenous neuroprotectors, anti-inflammatory agents, and stem cell and neuronal restoration agents, among others. Future research is needed to strengthen the evidence supporting clinical management and improving functional outcomes after TBI (Table 1 and Table 2).

## Figures and Tables

**Table 1 biomedicines-12-00781-t001:** Summary of clinical parameters for the management of TBI.

Clinical Parameters	Current Recommendations	Studies	Considerations
Cerebral spinal fluid (CSF) drainage	CSF drainage in patients with GCS < 6 within the first 12 h after injury to reduce ICP	Schizodimos et al. [36] Carney et al. [28]	There is limited evidence for the role of CSF drainage in management of elevated ICP in TBI.
Cerebrovascular hemodynamics	Maintain ICP below 22 mmHg CPP between 60 mmHg and 70 mmHg SBP ≥ 100 mmHg for patients 50–69 years old or ≥110 mmHg for patients 15–49 or >70 years old to decrease mortality and improve outcomes.	Schizodimos et al. [36] Carney et al. [28] Vella. et al. [41] Geeraerts et al. [33] Foundation B.T. [43]	There are two competing theories: - CPP of 60 mmHg with a target ICP < 20 mmHg; or - CPP > 70 mmHg with permissive ICP goal of 20–25 mmHg Recent data suggests there may not be a uniform CPP guideline for all patients. Individualized ranges of CPP may be the best route to optimize autoregulatory efficiency.
Decompressive craniectomy (DC)	DC is recommended to reduce mortality and improve neurologic outcomes in patients with severe TBI and late-refractory elevated ICP	Schizodimos et al. [36] Geeraerts et al. [33] Cooper et al. [51] Hutchinson et al. [52] Hawryluk et al. [53]	DC has been effective in reducing ICP and shortening ICU stay but has not been proven to improve long-term outcomes on Extended GOS. DC does not improve mortality in early-refractory ICP elevation. DC lowered 6-month mortality in late-refractory ICP elevation but showed higher rates of the occurrence of vegetative state and severe disability when compared to patients treated with medical therapy.
Barbiturates	High-dose barbiturates are recommended for increased ICP refractory to medical and surgical treatments. Barbiturates are not to be utilized as prophylaxis for intracranial hypertension.	Schizodimos et al. [36] Carney et al. [28] Vella et al. [41] Stocchetti et al. [74]	Barbiturates may be used to decrease ICP, decrease the cerebral metabolic rate of oxygen, and improve the overall oxygenation of cerebral tissues [74]. High-dose barbiturates significantly reduced ICP in patients with refractory ICP elevations but did not improve long-term death or disability compared to controls. The use of barbiturates to induce EEG burst suppression is not recommended for preventing the development of elevated ICP.
Glucose control	Guidelines for glycemic control suggest avoiding blood glucose < 100 and >180 mg/dL; insulin therapy should be instated if blood glucose > 180 mg/dL	Kurtz et al. [88] Ley et al. [92] Mowery et al. [93] Liu-DeRyke et al. [94] Vespa et al. [96] Hermanides et al. [97]	Hyperglycemia is correlated to injury severity and an independent predictor of mortality, and hypoglycemia worsens functional 6-month outcomes. The exact range for serum glucose levels in TBI patients has not been clearly determined, but it seems evident from the literature that serum glucose levels below 80 mg/dL and above 180 mg/dL are deleterious.
Anti-thrombolytics	TXA should be administered within 3 h of injury with known or suspected hemorrhage. TXA can be considered for patients with isolated TBI.	Zhang et al. [75] Roberts et al. [76] CRASH [77]	CRASH-3 trial differentiated data based on head injury severity showed the risk of head-injury-related death was significantly reduced in patients with mild–moderate TBI who received TXA compared to patients who received placebo. There was no significant risk reduction in patients with severe brain injury who received TXA.
Nutrition	Early nutrition therapy should aim for full caloric replacement by day 5 postinjury, or day 7 at the latest to decrease mortality. Nutrition therapy should be started within the first 24 h of injury.	Vizzini et al. [87] Kurtz et al. [88] Wang et al. [89] Cook et al. [90] Carney et al. [28]	Early feeding is associated with a significant reduction in mortality. Enteral nutrition can provide a better quality of micro- and macronutrients. Parenteral nutrition has classically been associated with higher risk of infection, immunosuppression, hyperglycemia, and hepatic steatosis compared to enteral nutrition. Trans gastric jejunal nutrition is preferred in ventilated patients to reduce the incidence of ventilator-associated pneumonia
Hyperventilation	Prophylactic hyperventilation during the first 24 hours after severe TBI should be avoided, as it can compromise cerebral perfusion. PaCO_2_ level < 35 mmHg rapidly reduces ICP.	Schizodimos et al. [36] Carney et al. [28]	Guidelines recommend hyperventilation as a temporary measure to reduce elevated ICP but it should be avoided within the first 24 h following the trauma as CBF is often reduced. Hyperventilation with partial pressure of arterial carbon dioxide (PaCO2) of ≤25 mm Hg is not recommended [28].
Osmotherapy	Hyperosmolar agents such as mannitol and hypertonic saline can reduce ICP via several proposed mechanisms, most notably an early hemodynamic effect and a late osmotic effect.	Schizodimos et al. [36] Vella et al. [41] Geeraerts et al. [33] Xiong et al. [70] Raslan et al. [71]	The gold standard ICP-lowering agent is mannitol, with serum osmolality maintained between 310 and 320 mOsm/L. Recent data have demonstrated hypertonic saline to be better at managing elevated ICP but did not show a difference in 6-month mortality or GOS compared to mannitol. Thus, current guidelines do not advocate for the use of a specific hyperosmolar agent.
Seizure control	Phenytoin is the gold standard recommendation to prevent early posttraumatic seizures. Currently, there is no anti-seizure medication recommended for late posttraumatic seizure prophylaxis.	Saletti et al. [105] Kruer et al. [108] Xu et al. [109] Temkin et al. [109] Zafar et al. [111]	Phenytoin can reduce the incidence of early seizures, but not late seizures. Recent studies have shown no statistically significant difference between phenytoin and levetiracetam in reducing early and late seizures, adverse effects, or mortality. Though levetiracetam has been associated with a better safety profile, there is not enough evidence to recommend the use of levetiracetam over phenytoin.
Temperature control	Body temperature should not exceed 37 °C. Early and aggressive temperature control measures such as IV and enteral antipyretic medication, control of room temperature, and cooling blankets should be initiated. Induced short-term prophylactic hypothermia is not recommended.	Schizodimos et al. [36] Carney et al. [28] Vella et al. [41] Bao et al. [82] Marion et al. [83] Hata et al. [84] Oddo et al. [85]	Treating fever with induced normothermia may lessen cerebral damage in patients with TBI. The application of induced normothermia is complicated by shivering, a physiologic thermoregulatory response. Shivering thus may limit the effectiveness of the therapeutic cooling meant to attenuate cerebral damage. There is conflicting evidence on the benefits of induced hypothermia in TBI patients. Risks of induced hypothermia seem to outweigh the benefits of early short-term prophylactic hypothermia.
Steroids	Steroids are not recommended for reducing ICP or improving long-term outcome in TBI patients.	Roberts et al. [113] Edwards et al. [114]	Trials showed an increased risk of death or severe disability in the group of patients who received steroids compared to controls.

**Table 2 biomedicines-12-00781-t002:** Summary of Experimental Investigations for management of TBI.

Experimental Investigations	Proposed Benefit	Studies	Considerations
Memantine	Memantine is a non-competitive NMDA open channel blocker. Thus, it may prevent glutamate excitotoxicity induced neuronal death. Investigations for use in TBI are currently in pre-clinical trials.	Rao et al. [116], Xia et al. [117], Marklund et al. [118]	In animal models of TBI, therapeutic doses of memantine can preferentially block extra synaptic NMDA receptors while sparing synaptic ones. Thus, memantine prevented neuron cell death in key hippocampal regions. Additionally, compared to pure NMDA receptor antagonists, memantine use was not associated with adverse effects.
Calcium channel blockers	Currently there are no guidelines for calcium channel blocker use in TBI patients. L-type and N-type calcium channel blockers can prevent cell damage caused by TBI via decreased intracellular Ca^+^ levels.	Kostron et al. [120] Xiong et al. [70] Langham et al. [121]	Despite their use in clinical practice to attempt to prevent secondary injury, a review of 6 randomized control trials demonstrated no statistically significant benefit of calcium channel blockers in reducing death and severe disability in TBI patients. However, there was a beneficial effect observed with nimodipine administration for patients with traumatic subarachnoid hemorrhage.
Nitric oxide	Investigations of Nitric Oxide use are currently confined to preclinical trials. Nitric oxide is a potent vasodilator and may be useful for treating reduced cerebral blood flow and ischemia after TBI.	Sawyer et al. [119]	Studies have shown improved outcome in patients with TBI treated with nitric oxide, which led to improved blood flow through the collateral circulation. However, caution is essential when administering nitric oxide as it may cause severe hypotension, leading to reduced CBF.
Statins	Statins have an anti-inflammatory effect via their role as HMG-CoA reductase inhibitors, and upregulate NO synthesis pathway, leading to improved capillary patency. Currently there are no guidelines for postinjury statin use.	Giannopoulos et al. [124] Jansen et al. [125]	Patients who used statins prior to injury were less likely to experience in-hospital mortality compared to non-users of statins. However, this positive effect was confined to patients on statins who did not have preinjury cardiovascular disease. This study also demonstrated better outcomes on GOS at 12 months in statin users compared to non-users of statins. However, there was no difference at 3 months postinjury.
S100B protein	S100B is a low molecular weight calcium binding protein released by glial cells following neuronal injury. Intraventricular S100B administration in preclinical trials demonstrated improved cognitive function and increased neurogenesis in the hippocampus.	Xiong et al. [70]	S100B has a dose dependent effect. At low concentrations, it is neuroprotective; however, at high concentrations, it can contribute to neuroinflammation. Since S100B can only enter the serum after blood brain barrier damage, S100B serum levels do not accurately predict intracranial S100B levels. However, monitoring serum S100B level can help predict patient outcome.
Erythropoietin	Exogenous erythropoietin may promote neuroprotection via binding to EPO receptors and exerting anti-excitotoxic, antioxidant, and anti-inflammatory effects. Currently, there are no guidelines regarding EPO use for TBI patients.	Ott et al. [127] Hasselblatt et al. [128] Brines et al. [132] Cerami et al. [133] Ponce et al. [134] Digicaylioglu, et al. [129] Jia et al. [130] Byts et al. [131]	EPO has known neuroprotective benefit; however, it is also a stimulator of erythropoiesis. Stimulated erythropoiesis following TBI may promote thromboembolic complications.
Caffeine	Upregulation of adenosine signal transduction is neuroprotective following TBI; caffeine is a nonselective adenosine receptor antagonist. Caffeine, therefore, may be neuroprotective following TBI.	Verrier et al. [135], Kochanek et al. [137], Sachse et al. [138]	Chronic caffeine exposure contributes to upregulation of Adenosine A1 receptor. The use of caffeine in TBI animal models has shown dose-and time- dependent variations in outcome. A recent observational study found a significant association between CSF caffeine > 194 ng/mL at admission and favorable GOS score at 6-months.
Cyclosporin	Following TBI, mitochondrial permeability transition pores (MPTP) are opened. Cyclosporin A can be neuroprotective following TBI because it inhibits MPTP opening.	Scheff et al. [140], Mazzeo et al. [141]	An infusion of Cyclosporin A in TBI patients, given within 12 hours of injury, posed no clinically significant safety concerns compared to a placebo infusion. However, patients who received the Cyclosporin A infusion did not have significantly different neurologic outcomes at 3- or 6-month post injury.
Anti-inflammatory therapies	Tumor necrosis factor (TNF) alpha plays a role in the inflammatory response that mediates secondary injury in TBI patients. Targeting microglial TNF alpha activation may improve outcomes in TBI patients.	Chio et al. [144], Baratz et al. [145]	When Etanercept, a TNF alpha antagonist, was administered to rats following TBI neurologic outcomes improved compared to animals given placebo. Similarly, 3, 6’-dithiothalidomide, a TNF alpha synthesis inhibitor, improved neurological outcomes in animals with induced mild TBI.
Thymosin B4	Thymosin B4 is a small peptide that sequesters G actin and is believed to play a role in neuronal regeneration.	Xiong et al. [146]	In animal models, high dose thymosin B4 treatment 6 hours post injury improved functional outcomes, reduced hippocampal cell loss, and improved neurogenesis.
Stem cell therapy	Stem cell interventions are proposed to enhance neurogenesis, angiogenesis, synaptogenesis, and immunoregulation by secreting chemokines and growth factors involved in recovery.	Luan et al. [151] Gennai et al. [152] Hu et al. [153] Tian et al. [154] Wang, et al. [155] Gritti et al. [156]	Preclinical studies have demonstrated the benefits of stem-cell-based therapies for brain injury. However, the mechanism of action of this intervention is unclear. Additional unanswered questions regarding safety, dose, route of administration, and timing of stem cell therapy after injury remain.

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
