# Peer review of "Clinical Management in Traumatic Brain Injury"

_biomedicines, 2024, doi:10.3390/biomedicines12040781_

Round 1

Reviewer 1 Report

Comments and Suggestions for Authors

The review article discusses TBI Clinical management. The authors have thoroughly reviewed the existing literature on TBI mechanisms, secondary TBI’s early management, and potential novel treatments. The article is well-written and provides existing clinical practices and future therapeutic potentials. Though I recommend this article for publication, there are a few comments and minor corrections in this article.

Minor comments:

This article mentioned focal and diffuse brain damage, but clinical management mainly focuses on focal damage.

Potential novel treatments, not mentioned, include recent progress using exosomes, non-coding RNAs, and stem cell treatment.

Minor Corrections:

Page 2, lines 95-96: “The text continues here” – what is it?

Page 6, line 286: The term PEEP is used for the first time; expand it.

Page 8, line 393: first 24h of injury – Throughout the manuscript, it is written as “hours,” but here, only “h” is used.

Page 9, line 454: TI as PTS – What is TI? Or is it TBI?

Page 11, line 550: The two words “isunclear” joined together—correct spelling. 

Author Response

Thank you for your edits, I have included the updated version (revisions highlighted)

Reviewer 2 Report

Comments and Suggestions for Authors

The review paper deals with brain trauma and its clinical treatment. The paper gives a good presentation of the mechanism of trauma, primary and secondary injury and clinical treatment. The following minor corrections are suggested to the authors:

 - Make sure that all used abbreviations are explained and correct tipfellers

 - Neuroinflammation (astrogliosis and activation of oligodendroglia and microglia) as well as the possibility of reactivation of dormant viruses are neglected in the mechanisms of secondary injury.

 - Methods of non-invasive therapy evaluation from blood markers are not mentioned

- The ineffectiveness of steroid therapy is explained, but not the mechanisms on which it should act- The positive sides of statin therapy are explained, but not the negative ones (cholesterol is every other cell membrane molecule, responsible for membrane fluidity and the production of neurites and myelinating cells)

- the role of myelin in preventing regeneration is omitted

- from chapter 4.10 one gets the impression that the proliferation of neurons is a significant process, but in the adult human brain its capacity is extremely small and refers only to interneurons in very limited regions of the brain

- considering all other regenerative processes that occur after trauma, which antidepressant therapy makes the most sense

Author Response

Thank you for your feedback, see attachment for edits. 

Reviewer 3 Report

Comments and Suggestions for Authors

The authors present a manuscript describing clinical management in Traumatic Brain Injury (TBI). Overall the conception and literature review is very comprehensive and written to describe multiple aspect of interventions. A few suggestions for editing:

-Overall the paper is describing clinical management for severe TBI ; although this is not stated in the intro which it needs to be. Currently there are multiple guidelines available for severe and mild TBI. The authors should indicate at the beginning of the paper that this clinical management is focused on severe TBI or if they want to include mild TBI should include a description of the guidelines and interventions that are related to this level of severity.

Into page 1 line 32- do you have a more recent reference for US data than 2005? It seems outdated.

Conclusion:

-You need to indicate in this if you are only discussing severe TBI.

-The summary of clinical guideline parameters should relate to severe TBI if this is the severity you are writing about.

It would be helpful to include some insights from your section on enriched environment in the conclusion section.  

Comments on the Quality of English Language

Please review and edit English. 

Author Response

Tnank you for your edits, please see attachment for revisions. 

Reviewer 4 Report

Comments and Suggestions for Authors

I appreciate the authors for presenting this review article. The content is well-organized. However, there are several articles on this topic, and most references date back to before 2018. I couldn't find anything particularly novel or distinctive. Therefore, my final decision is to reject it

Author Response

Thank you for your feedback, please see attachment for revisions

Reviewer 5 Report

Comments and Suggestions for Authors

This review covered various studis for TBI.

There are some typographical errors.

Author Response

Thank you for your feedback, please see attachment for revisions.
